# C1q Is Recognized as a Soluble Autoantigen by Anti-C1q Antibodies of Patients with Systemic Lupus Erythematosus

**DOI:** 10.3390/antib14040094

**Published:** 2025-11-05

**Authors:** Alexandra Anatolieva Atanasova, Ginka Ilieva Cholakova, Alexandra Panagiotis Kapogianni, Vancho Donev, Delina Ivanova, Anna Dimitrova Yordanova, Vanya Petkova Bogoeva, Ivanka Georgieva Tsacheva

**Affiliations:** 1Department of Biochemistry, Faculty of Biology, Sofia University “St. Kliment Ohridski”, 1164 Sofia, Bulgaria; alexandra.anatolieva@gmail.com (A.A.A.); ginka.cholakova@biofac.uni-sofia.bg (G.I.C.); kapojanni@uni-sofia.bg (A.P.K.); 2Bul Bio—NCIPD Ltd., 1504 Sofia, Bulgaria; vanedonev@abv.bg; 3Clinic of Rheumatology, UMHAT Sofiamed, 1797 Sofia, Bulgaria; delina_ivanov@abv.bg; 4Department of Molecular Biology of Cell Cycle, Institute of Molecular Biology “Rumen Tzanev”, Bulgarian Academy of Sciences, 1113 Sofia, Bulgaria; anna_gurova@bio21.bas.bg (A.D.Y.); vanya.bogoeva@gmail.com (V.P.B.)

**Keywords:** SLE, anti-C1q antibodies, soluble C1q

## Abstract

Background and Aims: C1q is an autoantigen in different autoimmune disorders, Systemic Lupus Erythematosus (SLE) and Lupus Nephritis (LN) among them. The two functional domains of C1q, the collagen-like region (CLR) and the globular head region (gC1q), are frequently recognized by autoantibodies in SLE and LN when C1q is immobilized. We studied whether autoantibodies to C1q in SLE and LN patients recognized C1q as a soluble autoantigen and whether the act of immobilization was a prerequisite for the recognition of C1q autoepitopes localized on gC1q domains. Methods: The interaction of soluble C1q and its globular fragments ghA, ghB, and ghC with immobilized IgG autoantibodies (and vice versa) from sera of 48 patients with SLE and LN was studied with ELISA. Data were compared using Spearman correlation coefficient. Fluorescence spectroscopy was used to study the interaction between C1q and LN IgG autoantibodies both presented in solution. Results: We found that anti-C1q autoantibodies from SLE and LN patients specifically bound C1q and gC1q fragments, ghA, ghB, and ghC, both as immobilized and soluble antigens. Correlation analysis indicated a negative correlation between the levels of autoantibodies against immobilized and soluble C1q and immobilized and soluble gC1q fragments which indicates different epitopes when these proteins were recognized as autoantigens in soluble and immobilized conformations. Conclusions: Serum C1q in patients with SLE is a target molecule for binding from anti-C1q autoantibodies. The gC1q region undergoes a conformational change in an immobilized and a soluble form, thus exposing different epitope-binding sites.

## 1. Introduction

The serum protein C1q is a key component of the complement system and it plays an important role in innate immune defense and in the appropriate development of adaptive immune response. C1q is a 460 kDa glycoprotein, consisting of 18 polypeptide chains of three types, designated A, B, and C. These chains form an N-terminal collagen-like region (CLR) and six C-terminal globular domains (gC1q), each representing a heterotrimer of globular head fragments named ghA, ghB, and ghC [1,2,3]. The complex structure of C1q determines its variety of functions. C1q is the ligand-binding molecule of the C1 complement component, which is a complex of C1q and the serine proteases C1r and C1s. The globular domains of C1q are involved in the recognition of a wide range of ligands and clearance of IgM- and IgG-containing immune complexes [4], apoptotic cells and cellular debris [5,6], in tissue remodeling during pregnancy [7,8,9], and in maintenance of immune tolerance [10,11,12]. Along with its crucial participation as a networking molecule between innate and adaptive immune response, C1q is also targeted as an autoantigen in many autoimmune disorders, including Systemic Lupus Erythematosus (SLE) and Lupus Nephritis (LN)—a common kidney disease, caused by SLE [13].

The autoantigenic features of C1q have been studied over a long period of time [14]. Soon after their discovery in 1971, anti-C1q antibodies were found to bind the CLR upon immobilization of C1q [15]. Since then, all research has been carried out assuming that the act of immobilization was a prerequisite for the binding of anti-C1q antibodies, which were found to result from conformational changes [16,17] of C1q as neo-epitopes or/and from post-translational modifications [18] or generated in response to proteolytic cleavage, protein activation, or protein binding [19]. The immobilization of C1q occurs upon binding to immune complexes or apoptotic cells [4,20,21]. The autoantibodies to C1q are a polyclonal fraction of the IgG class [22] with high affinity, which are a result of an antigen-driven process [23]. Vanhecke and coworkers identified, in 2012, the first linear epitope in CLR [24].

In 2007, we identified the immobilized ghA, ghB, and ghC of gC1q as targets for the SLE autoantibodies as well [25]. We further analyzed LN sera for anti-gC1q antibodies and we classified the gC1q autoepitopes as either neo-epitopes, exposed upon immobilization, or as conformational epitopes of the immobilized intact molecule, or as cryptic epitopes that become exposed by immobilized gC1q fragments [22].

Being a serum protein, C1q lacks intrinsic immunogenicity and it is likely that C1q is involved in the autoimmune condition rather than being an inducer. This assumption is supported by the fact that anti-C1q antibodies are not among the first autoantibodies to appear on the onset of SLE in contrast to anti-dsDNA antibodies, considered of highest sensitivity for SLE diagnosis [26]. Moreover, an anti-dsDNA epitope was mapped within the sequence of ghA and was suggested as a cross-epitope for the autoantibodies recognizing both dsDNA and C1q [27]. Along these lines, we hypothesized that there might be pre-existing antibodies, specific for epitope(s) presented by the conformation of the soluble C1q, which might or might not differ from the epitopes, exposed by the immobilized protein. We investigated whether C1q and its globular fragments ghA, ghB, and ghC were recognizable as soluble autoantigens by anti-C1q autoantibodies in patients with SLE and LN. In our previous work, we generated a structural analogue of a conformational globular autoepitope of C1q by selecting an anti-idiotypic single chain variable fragment (scFv) antibody, clone A1, which was found to exert a CDR structural homology to the apical region of gC1q [28].

## 2. Materials and Methods

### 2.1. Growth Media

The following growth media were used as follows: 50 × 5052 (25% glycerol, 2.5% glucose, 10% α-lactose); 20× NPS (66 g/L (NH_4_)_2_SO_4_; 136 g/L KH_2_PO_4_; 142 g/L Na_2_HPO_4_); ZYP medium for autoinduction (10 g/L Tryptone; 5 g/L Yeast Extract; 1 M MgSO_4_; 2% 50 × 5052; 5% 20× NPS; 100 μg/mL Ampicillin, pH 7.2); Luria–Bertani (LB) medium, pH 7.2 (10 g/L Tryptone, 5 g/L Yeast Extract, 10 g/L NaCl), supplemented with 1 mM MgSO_4_, 1% glucose, 100 μg/mL Ampicillin; and 5× M9 medium (1× M9, 0.2 mM MgSO_4_, 0.1 mM CaCl_2_, 0.2% glucose and 1.5% agar).

### 2.2. Buffers

PBS (pH 7.20): 0.137 M NaCl, 0.027 M KCl, 0.01 M Na_2_HPO_4_, 0.018 M KH_2_PO_4_; PBS 0.75: PBS containing 0.75 M NaCl; carbonate buffer (pH 9.60): 100 mM NaHCO_3_, 100 mM Na_2_CO_3_; TPBS: PBS containing 0.05% Tween-20; AP buffer (pH 9.60): 100 mM Tris, 100 mM NaCl, 5 mM MgCl_2_; phosphate buffer (pH 8.00): 0.1 mM Na_2_HPO_4_, 0.1 mM NaH_2_PO_4_, 0.5 M NaCl; and 2 M Imidazole (pH 8.00).

### 2.3. Patients

Sera from 48 Bulgarian patients with SLE, who attended the Nephrology Clinic of University Hospital “SofiaMed” over a period of one year from 10 December 2018 to 9 July 2019, were analyzed in the study. All patients provided written informed consent approved by the Ethics Commission of Sofia University. The cohort consisted of 44 females (91.67%) and 4 males (8.33%) with a median age of 52 years (range 24–75 years) and a median SLE duration of 6 years (range 2–8 years). At the time of blood sampling, the SLE patients were categorized according to EULAR/ACR SLE score as follows: all 48 patients were with moderate to severe Systemic Lupus Erythematosus, LN—10 patients, and NPSLE—9 patients, and 29 patients were with musculoskeletal, mucocutaneous, and another manifestation. All patients had SLEDAI-2K score ≥ 6 with at least 2 points for musculoskeletal or mucocutaneus manifestation at the time point of sampling.

Additionally, a pooled serum from 43 Bulgarian patients with previously diagnosed biopsy-proven LN who have attended, for a period of two and a half years, the Nephrology clinic of University Hospital “Queen Giovanna” of the Medical University, Sofia, was included in the study [28]. Pooled sera from 24 healthy donors were used as a control. All sera were stored at −30 °C.

### 2.4. Expression and Purification of Recombinant Proteins in E. coli

Expression of scFv A1 was induced by two alternative induction protocols as described previously [29]. Briefly, for IPTG induction, *E. coli* HB2151 cultures (OD_600nm_ of 0.8), containing a phagemid that encodes scFv clone A1 (*E. coli* HB2151/A1), were induced in LB growth medium for five hours at 25 °C with 0.5 mM IPTG. For autoinduction, overgrown *E. coli* HB2151/A1 daily cultures were inoculated in ZYP-autoinduction medium 1:100 and grown at 25 °C for 16 h. The induced cells were subjected to lysis in 1/20th of the culture volume with ice-cold lysis buffer 1 (100 mM Tris-HCl, pH 8.00) containing 20% sucrose (Fisher Scientific, Loughborough, UK) and 1 mM EDTA (Fisher Bioreagents, Fair Lawn, NJ, USA) for one hour on ice. Then, the lysate was centrifuged at 9000 rpm for 45 min at 4 °C, thus yielding supernatant (SN1), which contained soluble scFv. SN1 was kept on ice and later pooled with SN2, which was obtained after the subsequent lysis of the cell pellet in the same volume of ice-cold lysis buffer 2 (5 mM MgSO_4_) for 15 min on ice and then centrifuged for 45 min at 9000 rpm at 4 °C. The pooled supernatants were dialyzed overnight against Phosphate buffer (pH 8.00), containing 10 mM Imidazole (Acros Organics, Fair Lawn, NJ, USA) and applied on HIS-Select^®^ Ni-affinity gel column (Sigma-Aldrich, Saint Louis, MO, USA). The scFv A1 was eluted with Phosphate buffer (pH 8.00), containing 250 mM Imidazole. The eluted protein samples were dialyzed overnight in PBS (pH 7.20).

Expression of recombinant globular fragments ghA, ghB, and ghC was induced by both induction protocols as described above. They were expressed as fusion proteins with maltose-binding protein (MBP) in *E. coli* BL21(DE3) that were initially grown on M9 minimal media agar plates and then inoculated in the liquid growth media for induction of protein synthesis. Induced *E. coli* cells were lyzed in 50 mM Tris-HCl (pH 8.00), 0.5 M NaCl, 1 mM EDTA, 1 mM Benzamidine-HCl, 0.25% Tween-20, 0.25% Triton-x100, 0.25% NP-40, 100 μg/mL lysozyme for 15 min on ice. The lyzed cells were centrifuged at 9000 rpm for 45 min at 4 °C. The supernatants containing the recombinant globular head fragments were dialyzed overnight against 20 mM Tris-HCl (pH 8.0), 0.1 M NaCl, 1 mM EDTA, 5% Glycerol, and 0.25% Tween-20 and then purified by affinity chromatography on amylose resin (New England Biolabs, Ipswich, MA, USA). They were eluted with 20 mM Tris-HCl, pH 8.0, 0.1 M NaCl, and 10 mM Maltose. The purified proteins were dialyzed overnight in PBS (pH 7.20).

### 2.5. Biotin Labelling

C1q (Merck Millipore Calbiochem™ Calbiochem, Darmstadt, Germany) and its purified ghA, ghB, and ghC in PBS (pH 8.5) were each incubated with (+)-Biotin N-hydroxysuccinimide ester (10 μg/μL in DMSO, Acros organic, Saddle Brook, NJ, USA) at 1/6th of the amount of the protein sample for 4 h at room temperature on a shaker. After biotinylation, the proteins were dialyzed against PBS (pH 7.20), overnight at 4 °C.

### 2.6. ELISA Assays

In all ELISA experiments, the blocking step was performed with 1% BSA (bovine serum albumin, Sigma) 200 μL/well for 1 h at 37 °C. After every incubation, the plates were washed 3 times with TPBS (200 μL/well). All samples were analyzed in triplicate.

#### 2.6.1. ELISA Assay for the Recognition of Soluble C1q and Its Globular Head Fragments ghA, ghB, and ghC by Immobilized IgG Autoantibodies from Sera of Patients with LN

IgG_LN_ fraction (2.85 µg/µL), purified by affinity chromatography on Protein G from pooled LN patients’ sera, was diluted in 1:2 ratio in carbonate buffer (pH 9.60) and immobilized on a 96-well microtiter plate overnight at 4 °C. After blocking, a biotinylated C1q or biotinylated ghA, ghB, or ghC were incubated in increasing amounts of 0.125–0.25–0.5–1–2–4 μg/well in PBS 0.75 overnight at 4 °C. The formed complexes were detected by Extravidin-AP (Sigma-Aldrich, St. Louis, MO, USA) and pNPP (p-Nitrophenylphosphate, Life Technologies, Inc., Gaithersburg, MD, USA) diluted in AP buffer. The absorbance was read at 405 nm using a DR-200B Microplate Reader (Wuxi Hiwell Diatek Instruments Co., Wuxi, China).

#### 2.6.2. Competitive ELISA

IgG_LN_ was diluted in 1:2 ratio in carbonate buffer (pH 9.60) and immobilized on a 96-well microtiter plate overnight at 4 °C. A biotinylated C1q (2 μg/well) was pre-incubated with the tested competitors ghA, ghB, ghC, and scFv clone A1 in increasing amounts of 2.5–5–10 μg/well in PBS 0.75 for 20 min at room temperature and then transferred onto the IgG-coated plates. The following incubating steps and reagents are the same as mentioned above. The formed complexes were detected as described above.

#### 2.6.3. ELISA Assay for the Recognition of C1q, ghA, ghB, and ghC by Sera from SLE Patients

This ELISA assay was performed in two settings. In one setting, the four test antigens were immobilized (2 µg/well in carbonate buffer) and incubated with sera from 48 patients with active SLE (diluted 1:50 in PBS 0.75, 100 μL/well) overnight at 4 °C. The formed complexes were detected by goat anti-human IgG-AP and pNPP. In the alternative setting, the plates were coated with 48 SLE sera (diluted 1:50 in carbonate buffer, 100 μL/well) and incubated with biotinylated soluble C1q, ghA, ghB, or ghC (2 μg/well in PBS 0.75) overnight at 4 °C. The formed complexes were detected by Extravidin-AP and pNPP.

### 2.7. Fluorescent Spectroscopy

Binding of IgG_LN_ with soluble C1q was analyzed by a fluorescence titration method applicable for its intrinsic sensitivity. Fluorescence spectra were recorded with a Shimadzu RF-5000 spectrofluorometer (Kyoto, Japan), equipped with a 10 mm quartz cuvette holder. Measurements were performed using a fluorescent dye ANS (8-Anilino-1-naphthalenesulfonic acid), with a slit width of 10, 15 nm. Concentration of ANS was calculated spectrophotometrically. In order to measure precisely the fluorescence intensity for each of the ligand concentrations, necessary for the formation of IgG_LN_-C1q complexes, a 1 min equilibration period was used. Fluorescence intensity was corrected for dilution, respectively. The maximal increase in the ANS fluorescence due to the saturation of the binding sites by the ligand (Fmax) was evaluated from the experimental data. In all experiments, the absorbance of the samples at the excitation wavelength was kept less than 0.05 in order to minimize inner filter and self-absorption effects. All fluorescence measurements were carried out in PBS 0.75 at 23 °C.

### 2.8. Statistical Analysis

Statistical analysis of results was performed in GraphPad Prism software version 8.0.1. Mann–Whitney U test for continuous variables for 2-group comparisons was used. Quantitative data were expressed as mean ± SD. The Spearman correlation was used to analyze correlation. Statistical significance was considered as *p* < 0.05.

## 3. Results

### 3.1. Clinical Evaluation of Patients

After diagnosis, the frequency of symptoms such as butterfly rash, arthritis, and hematological abnormalities remained stable, whereas the prevalence of central nervous manifestations increased from 8 to 60%, with renal involvement up to 43%.

Examples of frequently observed skin manifestations were livedo reticularis (20%) and digital skin vasculitis (10%); other clinical signs such as urticaria (3%), panniculitis (1%). Regarding the cardiovascular and pulmonary system, the most frequent clinical signs were Raynaud’s phenomenon (52%), hypertension (43%), pleuritis (10%) and pericarditis (8%), whereas myocarditis was only observed in 4% of the patients. Anemia was diagnosed at least once during the disease course in 28% of the patients. There were no cases with hemolytic anemia, and in 24% of the cases, it was an anemia of chronic disease. Leukocytopenia was observed in 28% and thrombocytopenia in 24% of the patients. The most frequent clinical signs related to abdominal abnormalities were the findings of hepatomegaly (10%) and splenomegaly (6%). Other symptoms, such as pancreatitis and intestinal vasculitis, were not registered in this group of patients.

### 3.2. Fluorescent Analysis

We studied the interaction of affinity-purified IgG autoantibodies from patients with LN (IgG_LN_) to soluble C1q using ANS fluorescence. ANS was excited at 375 nm (Figure 1). C1q was titrated with increasing concentrations from 0 to 20 μmol C1q in PBS 0.75. The binding of ANS to the complex caused enhancement of the extrinsic ANS fluorescence. The experimental data were fitted to a hyperbolic curve, showing a high-affinity binding site. The apparent dissociation constant was calculated, K_D_ = 4.9 μM, which showed high affinity binding of the complexes.

### 3.3. ELISA Analyses

Alternatively, in contrast to the experimental conditions of the fluorescent analysis where both IgG autoantibodies and C1q were soluble, we studied the same interaction by ELISA where IgG_LN_ were immobilized and incubated with soluble C1q in the presence of 0.75 M NaCl. The high ionic strength was applied in order to inhibit the binding of C1q to CH_2_ domains of IgG molecules with irrelevant antigenic specificities. We registered a dose-dependent binding between the immobilized IgG_LN_ and the soluble C1q (Figure 2A). The high signal of the ELISA assay supported the estimated high affinity of the interaction based on the fluorescent data.

Further on, we analyzed the recognition of ghA, ghB, and ghC presented in soluble form by the immobilized IgG_LN_. We observed a dose-dependent interaction between IgG_LN_ and soluble globular head fragments with the highest signal produced by ghC and generally weak binding by ghA and by ghB (Figure 2B). The weaker binding of the immobilized IgG_LN_ to soluble ghA and ghB could be related either to the absence of exposed autoepitopes on these globular head fragments when presented in a soluble form or due to their participation in conformational autoepitope and insufficient affinity as single molecules to form detectable amounts of immune complexes. In order to verify their contribution to the formation of globular autoepitope of the soluble C1q, we performed a competitive ELISA, in which every globular fragment was analyzed for its inhibitory capacity on the recognition of soluble C1q by the immobilized IgG_LN_ autoantibodies. In the competitive analysis we also included scFv A1 as a structural analogue of the localized conformational globular autoepitope of C1q. All studied proteins showed approximately 50% inhibitory capacity suggesting that all globular head fragments of soluble C1q participated in composing of an autoepitope localized in the apical part of gC1q (Figure 2C).

Next, we analyzed the recognition of both conformational states of C1q in additional 48 sera from patients with active SLE in order to assess statistically the frequency and distribution of anti-soluble C1q compared to anti-immobilized C1q among individuals with SLE. First, we detected both types of anti-C1q antibodies in 24 sera of randomly selected healthy donors. The antibodies to the immobilized C1q had a mean value of 0.142 (average 0.148), determined as OD_405nm_, an interquartile range from 0.080 to 0.170, and range from 0.041 to 0.382. The antibodies to the soluble C1q had a mean value 0.073 (average 0.0735), determined as OD_405nm_, an interquartile range from 0.037 to 0.088, and range from 0.018 to 0.172 (Appendix A). The SLE sera were considered positive for autoantibodies if their value exceeded the cut-off value, calculated by the sum of the average and the S.D. values of the control, represented by the pooled serum of the 24 healthy donors. We found that the positive patients for autoantibodies against immobilized C1q, ghA, ghB, and ghC were 31.25% (15/48), 16.66% (8/48), 8.33% (4/48), and 8.33% (4/48), respectively (Figure 3A and Table 1). The frequencies of positive patients with antibodies against soluble C1q, ghA, ghB, and ghC were 6.25% (3/48), 33.33% (16/48), 31.25% (15/48), and 8.33% (4/48), respectively (Figure 3B and Table 1). Autoantibodies from the SLE patients recognized, with significantly higher affinity, the immobilized C1q than the soluble C1q (*p* = 0.0027) (Figure 3C), but there was no significant difference of binding affinity of the autoantibodies against immobilized and soluble ghA, ghB, and ghC (Figure 3D–F). The screening revealed that C1q was recognized only as a soluble antigen in 2.08% (1/48) sera and, in 27.08% (13/48) of the serum samples, was bound only as immobilized antigen (Table 1). In 4.17% (2/48), the protein was recognized in both conformational states—immobilized and soluble. Interestingly, the majority of the positive anti-immobilized C1q (12/15, 80%) contained antibodies to one or more soluble globular head fragments. All sera that were found positive for anti-soluble C1q also contained autoantibodies recognizing at least one globular fragment.

The soluble globular head fragments, one of them or in a combination, were recognized by antibodies in 54.17% (26/48) of the tested sera. The ghA was recognized only as a soluble antigen in 25% (12/48) of the samples and only as immobilized antigen in 8.33% (4/48) of the tested sera. The ghB was bound only in soluble form by the SLE autoantibodies in 29.17% (14/48) of the tested sera and only as immobilized antigen in 6.25% (3/48) of the samples. The weakest antigenic binding was found for ghC—6.25% (3/48) only in soluble conformation and 6.25% (3/48) only in immobilized conformation.

The binding affinity of the autoantibodies to immobilized C1q negatively correlated with those to soluble C1q (r = −0.021, *p* = 0.885) (Figure 4A). Such negative correlation trend was also observed when we analyzed the binding affinity of autoantibodies to immobilized and soluble ghA (r = −0.036, *p* = 0.809) and ghC (r = −0.079, *p* = 0.594) (Figure 4B,D). The analysis showed a moderate correlation between the binding affinity of autoantibodies that recognize immobilized ghB and soluble ghB (r = 0.406, *p* = 0.004) (Figure 4C).

## 4. Discussion

The autoimmune disease SLE and its common kidney manifestation LN are associated with the pathological presence of autoantibodies against complement C1q. So far, it is well estimated that anti-C1q autoantibodies primarily bind to epitopes exposed on a ligand-bound immobilized C1q due to a conformational change that occurs within the collagen-like region of the C1q and these autoantibodies do not or very weakly bind to soluble C1q [13,15,18]. Our study started with analysis of IgG, isolated from pooled sera of LN patients who have been subjected to two and a half years of follow-up [22]. The LN manifestation of SLE was chosen because it is known to be marked by the presence of anti-C1q antibodies. The epitope specificities of the antibodies to immobilized C1q presented them as a dynamic population which targeted a whole array of epitopes, not just CLR epitopes. In the present study, we aimed at detecting generally unknown autoepitopes to the soluble C1q, so we tested pooled LN sera as a source of all possible C1q autoepitopes including less represented ones which required a very sensitive technique such as fluorescent spectroscopy (Figure 1) and competitive ELISA (Figure 2). In that way we were able to demonstrate, for the first time, that soluble C1q as a serum molecule is also a target for the autoimmune antibodies most likely by binding epitopes within its gC1q. The dose-dependent trend of the interaction registered by the highly sensitive fluorescent technique and supported by the immunosorbent analyses defined it as specific and of high affinity. The interaction was performed in high ionic strength in order to rule out the possibility of C1q binding to Fc fragments of irrelevant IgG specificities. Additionally, in our ELISA system, the soluble antigens were biotinylated. As the SLE autoantibodies are of IgG class, the binding of C1q to Fcγ is known to involve ghB with its key Arg114 amino acid [30,31]. We assume that the process of biotinylation most probably hinders Arg114 (Figure 5) by the surrounding modified Lys residues and, as a result, the interaction is inhibited. Consequently, the biotinylation would strongly favor the possibility for C1q to bind IgG molecules which were C1q-specific because the interaction would be via Fab.

After we established the soluble C1q as a target for antibodies in the pooled LN serum, we needed to verify it for SLE patients with a different set of clinical symptoms, not only renal (Figure 3). Analyzing the individual patients’ sera helped us assess, statistically, the frequency and distribution of anti-soluble C1q compared to anti-immobilized C1q antibodies, and also to distinguish the epitopes targeted on the soluble C1q from those on immobilized C1q. The most differences were found for the globular head fragments of gC1q, known antigens for the autoantibodies in immobilized state [14]. In the present study, we found a different pattern of recognition for gC1q suggesting different epitope(s) presented by the conformation of the soluble fragments (Table 1). Sera, which were negative for antibodies to the immobilized globular head fragments, were found positive for antibodies to the soluble state of the same proteins in 37.5% (18/48) of the analyzed sera. The correlation analysis showed a negative correlation coefficient between the presence of autoantibodies against soluble and immobilized C1q, ghA, and ghC which supported the suggestion of different epitopes when these proteins were recognized as autoantigens in soluble and immobilized conformations. Interestingly, the correlation between anti-ghB autoantibodies against soluble and immobilized ghB was estimated as moderate presumably due to similar autoepitopes presented in both conformational states. Future experiments are needed to determine at which time point of the disease course anti-soluble C1q autoantibodies appear and what the structural features of the epitopes are that are localized on the globular head fragments ghA, ghB, and ghC when C1q is not immobilized on a solid surface.

The observed higher numbers of positive sera for anti-soluble ghA and anti-soluble ghB in comparison with their recognition as immobilized antigens suggests that they are the main target of the anti-soluble antibodies against C1q. Such antibodies are likely to impair the apoptotic clearance function of C1q. These two globular head fragments are the key constituents of the phosphatidylserine-binding site of C1q [33] and consequently for the binding of apoptotic blebs [34]. This new aspect of C1q autoantigenicity could explain the participation of C1q in the vicious circle of breaking the immune tolerance by autoantigen release from unattended apoptotic cells according to the “waste disposal” hypothesis [35], thus fueling the autoimmune response. Moreover, ghA contains the cross-autoepitope, reported by Franchin et al. [27], and mimicked structurally by scFv A1 [28], suggesting that the already existing anti-dsDNA antibodies could be a contributing factor for the involvement of C1q in the autoimmune setting of SLE.

Anti-C1q antibodies are not exclusive for SLE. Indeed, C1q was first found to be an autoantigen in SLE but other autoimmune disorders are also characterized by anti-C1q antibodies like rheumatoid arthritis, mixed connective tissue disease, Felty’s syndrome, rheumatoid vasculitis, polyarteritis nodosa, polychondritis, Sjögren’s syndrome, etc. Interestingly, the clinical symptoms of the SLE cohort, analyzed in this study, were diverse (Table 1) and included rheumatoid arthritis, vasculitis, and Sjögren’s syndrome. However, we did not find an association of a symptom or a combination of symptoms with any epitope specificity of the soluble autoantigen C1q. The cutaneous and joint symptoms were ubiquitous. Patients presenting with LN (sera 3, 4, 9, 23, and 42) were positive for both anti-immobilized and anti-soluble epitopes of C1q. ANA antibodies are the hallmark of SLE and are used for diagnosing the condition. In this SLE cohort, 16/48 (33.3%) sera were positive for ANA (sera 2, 3, 4, 5, 7, 8, 10, 11, 17, 21, 24, 35, 38, 40, 43, and 45). In five of them (5/48, 10%), we detected antibodies only to soluble ghA (sera 2, 4, 7, 8, and 11). Some of the ANA positive sera (6/48, 12.5%) also had anti-immobilized C1q antibodies, alone (serum 24) or in combination with anti-soluble ghB (sera 21, 38, and 45), or in combination with anti-soluble C1q, anti-soluble ghA, and anti-soluble ghB (serum 40), or in combination with anti-soluble ghC (serum 17). This diverse spectrum implies a personalized reactivity to the protein among SLE patients. This is hardly unexpected given the complex conformation of such a large molecule as C1q and it opens up the following question as to whether the anti-soluble C1q antibodies are the trigger of the conformational change turning CLR into the antigenic part of C1q, given they are increased above the physiological threshold.

## 5. Conclusions

We found that both immobilized and soluble C1q were targeted by SLE and LN autoantibodies and that the autoepitopes of the two conformations were different. The soluble protein was bound by epitopes localized in gC1q mainly through ghA and ghB. The anti-soluble gC1q antibodies might be a contributing factor for SLE progression most likely by interfering with the apoptotic clearance function of C1q.

## Figures and Tables

**Figure 1 antibodies-14-00094-f001:**
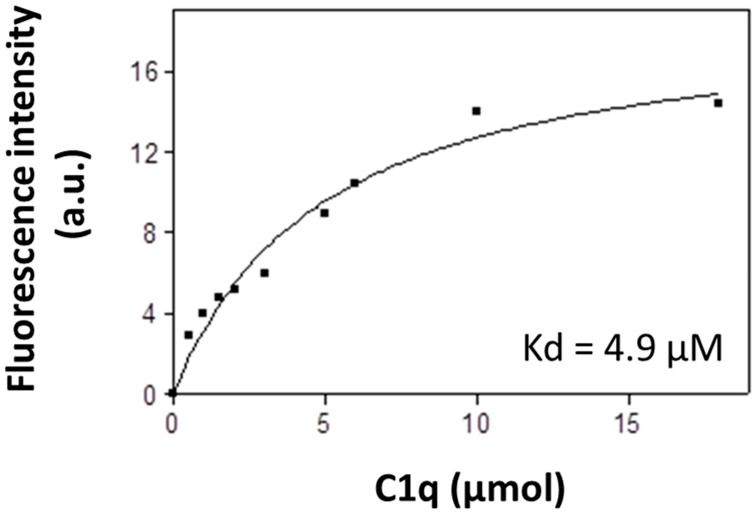
A soluble C1q is targeted by IgG autoantibodies from patients with LN. Hyperbolic binding curve representing the binding of IgG autoantibodies from patients with LN with soluble C1q. Data were analyzed by a nonlinear regression for a single binding site, showing the theoretical best fit to the experimental data; a.u., arbitrary units.

**Figure 2 antibodies-14-00094-f002:**
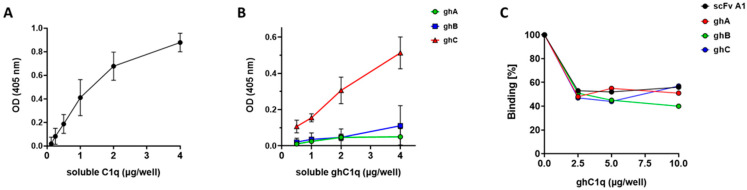
An autoepitope on soluble C1q is localized in the apical part of gC1q. (**A**) Dose-dependent interaction of immobilized IgG_LN_ autoantibodies and a soluble C1q. (**B**) Interaction of immobilized IgG_LN_ autoantibodies and soluble globular head fragments. Error bars: mean ± SD. (**C**) Competitive analysis of the interaction between immobilized IgG_LN_ and soluble C1q with ghA, ghB, ghC, and scFv A1.

**Figure 3 antibodies-14-00094-f003:**
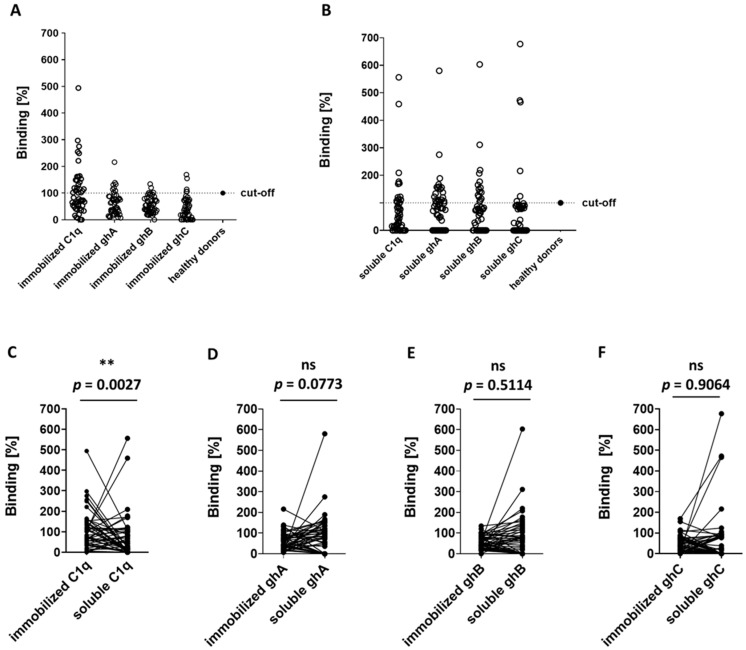
Presence of autoantibodies to immobilized or soluble C1q and gC1q fragments in sera of SLE patients. Autoantibodies to immobilized C1q (panel (**A**)) or soluble C1q (panel (**B**)) and gC1q fragments in sera of SLE patients. One dot = one individual serum; control—pooled sera from healthy donors (*n* = 24). Panels (**C**,**D**): Comparative analysis of anti-C1q (panel (**C**)), anti-ghA (panel (**D**)), anti-ghB (panel (**E**)), and anti-ghC (panel (**F**)) autoantibodies to immobilized and soluble C1q and gC1q in individual SLE sera. One dot = one serum sample, Mann–Whitney u-test, significance is denoted by ** (*p* ≤ 0.01), ns—nonsignificant.

**Figure 4 antibodies-14-00094-f004:**
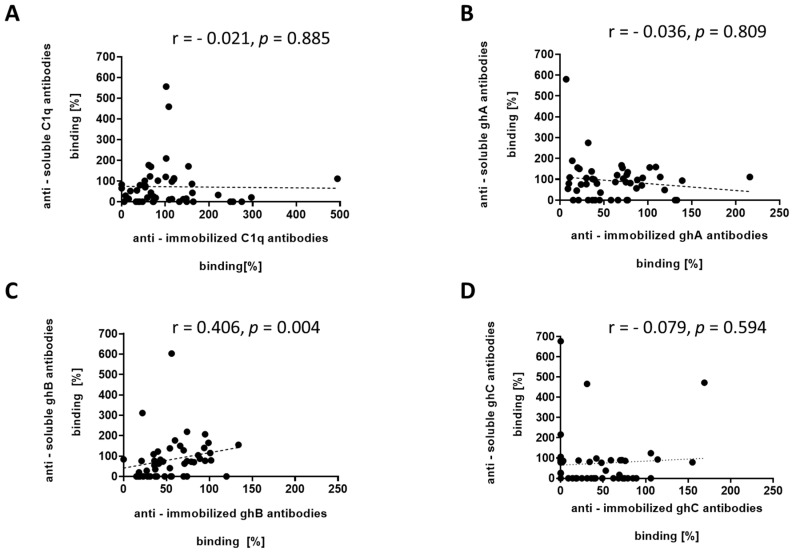
Correlation analysis. Correlation between the levels of autoantibodies against immobilized and soluble C1q (panel (**A**)) and immobilized and soluble gC1q fragments (panel (**B**–**D**)). One dot = one patient, Spearman correlation.

**Figure 5 antibodies-14-00094-f005:**
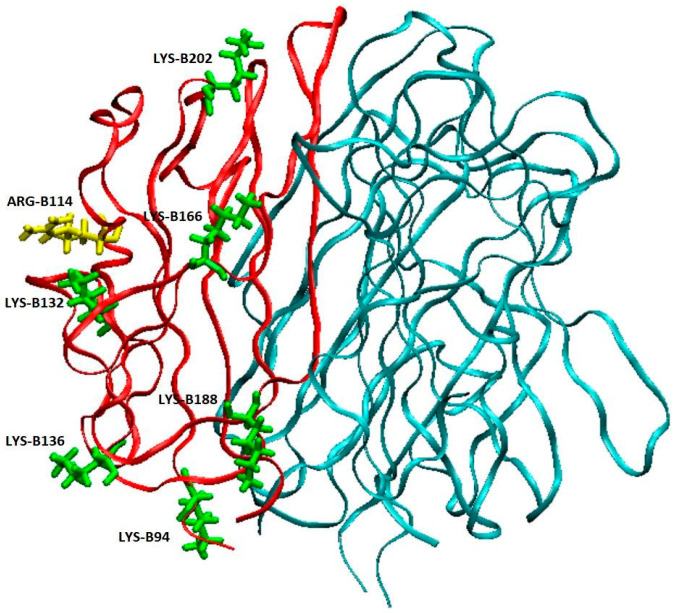
Schematic representation of the structure of human gC1q. The globular heads ghA and ghC are shown in cyan, and ghB in red, with Arg114 (yellow), the key residue for the interaction of C1q with Fc of IgG, and proximal Lys residues (green). VMD 1.9.4 [32] is used for visualization.

**Table 1 antibodies-14-00094-t001:** Positive individual SLE sera for the presence of autoantibodies against immobilized and soluble C1q and its globular head fragments.

Serum Number	Immobilized				Fluid-Phase				Clinical Symptoms
	C1q	ghA	ghB	ghC	C1q	ghA	ghB	ghC	
1						+			cutaneous, joint, vascular, cerebrovasculitis
2						+			cutaneous, joint, ocular–pulmonary, symptomatic myositis, minimum 1:80 ANA antibodies
3		+				+			cutaneous, joint, renal, minimum 1:80 ANA antibodies
4						+			cutaneous, joint, Lupus Nephritis, minimum 1:80 ANA antibodies
5		+							cutaneous, joint, minimum 1:80 ANA antibodies
7						+			cutaneous, joint, Necrotic vasculitis in Raynaud’s syndrome, minimum 1:80 ANA antibodies
8						+			cutaneous, joint, minimum 1:80 ANA antibodies
9		+			+	+			cutaneous, joint, Lupus Nephritis 3 degree
10				+					cutaneous, joint, cerebrovasculitis, minimum 1:80 ANA antibodies
11						+			cutaneous, joint, minimum 1:80 ANA antibodies
13							+		cutaneous, joint, vascular, myopathy
14						+			cutaneous, joint, cardiovascular with rhythm disorders
15				+		+	+		cutaneous, joint, vascular
16	+			+			+	+	Rheumatoid polyarthritis—affecting joints of the hands and feet, elbow and shoulder joints
17	+							+	cutaneous, joint, minimum 1:80 ANA antibodies
18		+							seropositive RA with high joint count
19						+	+		seropositive RA with high joint count
21	+						+		cutaneous, joint, minimum 1:80 ANA antibodies
22							+		seropositive RA affecting small joints of the hands and feet, shoulder, and knee joints
23	+								cutaneous, joint, Lupus Nephritis 1 degree
24	+								cutaneous, joint, Necrotic vasculitis in Raynaud’s syndrome, central nervous system, minimum 1:80 ANA antibodies
25							+		cutaneous, joint, vascular, Raynaud’s syndrome
29		+		+					cutaneous, joint, vascular, cerebrovasculitis
35		+	+						cutaneous, joint, vascular, minimum 1:80 ANA antibodies
36	+						+		seropositive RA affecting the joints of the hands, feet, and knees
37	+		+					+	cutaneous, joint, vascular, Raynaud’s syndrome
38	+						+		cutaneous, joint, minimum 1:80 ANA antibodies
39	+					+	+		cutaneous, joint, myopathy, cerebrovasculitis, multifocal encephalopathy
40	+				+	+	+		cutaneous, joint, cerebral artery aneurysm, minimum 1:80 ANA antibodies
41	+	+	+		+	+	+		seropositive RA and secondary Sjögren’s syndrome
42	+					+	+	+	cutaneous, joint, Lupus Nephritis 2 grade
43			+						cutaneous, joint, minimum 1:80 ANA antibodies
45	+						+		cutaneous, joint, vascular, minimum 1:80 ANA antibodies
46	+	+				+	+		seropositive RA with high joint count
47	+								cutaneous, joint, vascular

Positive sera for any autoantigen, immobilized or soluble, are designated by “+’’.

## Data Availability

The original contributions presented in this study are included in the article/Appendix A. Further inquiries can be directed to the corresponding author(s).

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
