# Peer review of "C1q Is Recognized as a Soluble Autoantigen by Anti-C1q Antibodies of Patients with Systemic Lupus Erythematosus"

_2073-4468, 2025, doi:10.3390/antib14040094_

Round 1
Reviewer 1 Report
Comments and Suggestions for Authors
The study investigates whether anti-C1q autoantibodies from patients with systemic lupus erythematosus (SLE) and lupus nephritis (LN) recognize soluble C1q in addition to the traditionally studied immobilized form. Using ELISA and fluorescence spectroscopy, the authors show that patient-derived antibodies bind both soluble C1q and its globular fragments (ghA, ghB, ghC), albeit with distinct epitope preferences compared to immobilized C1q. They report that soluble gC1q, especially ghA and ghB, is a frequent target of autoantibodies in SLE sera. These findings suggest that conformational differences between soluble and immobilized C1q expose different epitopes, potentially contributing to defective apoptotic clearance and disease progression in SLE.
Major Issues to address:
-The fluorescence titration experiment (Figure 1) was performed only with IgG purified from LN patients. In contrast, Figures 2–3 include analyses of the entire SLE cohort (n=48). This inconsistency makes cross-comparison misleading. The text should explicitly state that Figure 1 represents LN data only and clarify whether results are generalizable to SLE.
-Figure 3. The cut-off for positivity is defined as mean + SD of healthy controls (n=24). However, the distribution of antibody levels in healthy donors is not shown. This makes it difficult to judge whether healthy values overlap with patients. Recommendation: add a panel showing the distribution for healthy donors, or at least explicitly report median, IQR, and range.
-The study includes 48 SLE patients (10 LN, 9 NPSLE, 29 others) but also a pooled LN cohort (n=43) from another hospital. This mixing of cohorts (individual vs pooled) complicates interpretation and may bias the analysis. Authors should justify this design and clearly separate results from pooled vs individual sera. Also, make a table with detailed patients’ clinical characteristics.
-Discussion states biotinylation of lysines “hinders Arg114” and prevents Fc binding (Figure 5). This is speculative without experimental evidence. This statement should be toned down or supported with structural/mutational references.
Minor Issue
Patient sex distribution is described as “44 females (98.08%) and 4 males (1.92%)”: the percentages are incorrect (should be 91.7% vs 8.3%).
Author Response
Please, see the attachment.

Reviewer 2 Report
Comments and Suggestions for Authors
Dear authors, the manuscript is well-written and easy to follow. I extend my congratulations for what it seems was a difficult task.
However, I would like to suggest that you expand the discussion a little further, for example, by detailing the potential clinical implications of C1q whether immobilized or soluble, as well as explaining the importance of the epitopes that bind to gC1q being ghA and ghB, rather than ghC, and by discussing the clinical characteristics of the patients in more detail.
Author Response
Please, see the attachment.

Round 2
Reviewer 1 Report
Comments and Suggestions for Authors
The major issues have been addressed